# Bandpass Filter Integrated Metalens Based on Electromagnetically Induced Transparency

**DOI:** 10.3390/nano12132282

**Published:** 2022-07-02

**Authors:** Dongzhi Shan, Jinsong Gao, Nianxi Xu, Hai Liu, Naitao Song, Qiao Sun, Yi Zhao, Yang Tang, Yansong Wang, Xiaoguo Feng, Xin Chen

**Affiliations:** 1Key Laboratory of Optical System Advanced Manufacturing Technology, Changchun Institute of Optics, Fine Mechanics and Physics, Chinese Academy of Sciences, Changchun 130033, China; shandongzhi@ciomp.ac.cn (D.S.); xnxlzhy999@126.com (N.X.); nmliuhai@163.com (H.L.); songnaitao@ciomp.ac.cn (N.S.); sqsunqiao@126.com (Q.S.); zyy15555157812@126.com (Y.Z.); tangyang0816@163.com (Y.T.); wangyansong99@163.com (Y.W.); fxg74@163.com (X.F.); chenxin_19344834@163.com (X.C.); 2University of the Chinese Academy of Sciences, Beijing 100039, China; 3Jilin Provincial Key Laboratory of Advanced Optoelectronic Equipment and Instrument Manufacturing Technology, Changchun 130033, China

**Keywords:** metalens, electromagnetically induced transparency metasurface, bandpass filter, long-wavelength infrared, integrated devices

## Abstract

A bandpass filter integrated metalens based on electromagnetically induced transparency (EIT) for long-wavelength infrared (LWIR) imaging is designed in this paper. The bandwidth of the metalens, which is a diffractive optical element, decreases significantly with the increase of the aperture size to a fixed f-number, which leads to the decline of imaging performance. The same material composition and preparation process of the metalens and the EIT metasurface in the long-wavelength infrared make it feasible that the abilities of focusing imaging and filtering are integrated into a metasurface device. With the purpose of validating the feasibility of this design method, we have designed a 300-μm-diameter integrated metalens whose f-number is 0.8 and the simulation was carried out. The introduction of EIT metasurface does not affect the focusing near the diffraction limit at the target wavelength, and greatly reduces the influence of stray light caused by non-target wavelength incident light. This bandpass filter integrated metalens design method may have a great potential in the field of LWIR compact optical systems.

## 1. Introduction

The long-wavelength infrared (LWIR) imaging technology attracts more and more attention due to its great potential in many important areas, such as facilities maintenance and environmental monitoring [1], and hopefully the LWIR lens, which is the core device in the optical imaging system, can be developed to meet the lightweight and compact assembly requirements [2]. Metalens, as a metasurface device composed of subwavelength electromagnetic scatterers, is regarded as a potential option in the compact optical system due to it being able to modulate the wavefront with sub-wavelength spatial resolution [3,4,5,6,7,8] and its ultra-thin thickness [9,10]. Therefore, some excellent research results of the LWIR metalens have been presented [11,12,13].

Different from the refractive lens, the metalens achieves the focusing effect based on the diffraction effect instead of the result of the equivalent optical paths [14], and this feature results in an obvious chromatic aberration caused by the diffraction effect with the increase of the aperture size of the metalens to a fixed f-number. To improve that, some excellent research results of the achromatic metalens have emerged [15,16,17,18]. However, the finite range of the refractive index of the material used in the LWIR imaging systems leads to a mutual restriction relationship between the numerical aperture (NA) and the aperture size of the achromatic metalens [17,19], which makes it quite difficult to achieve a high-resolution imaging performance for the achromatic metalens with a large aperture size. Therefore, some realizable design methods need further exploration for the LWIR metalens with a large aperture size to achieve a high-quality imaging performance.

Within this paper, we come up with a strategy to design a bandpass filter integrated metalens based on electromagnetically induced transparency (EIT) to circumvent the chromatic aberration by eliminating the influence of the stray light for LWIR imaging. First, the design method of the integrated the metalens was illustrated after the chromatic aberration of the diffractive metalens was analyzed. Then, the significance of integrating the EIT metasurface with the metalens is discussed since the material composition and preparation process are consistent. Finally, in order to demonstrate the feasibility of this design method, a bandpass filter integrated metalens with a diameter of 0.30 mm and an f-number of 0.8 has been designed to operate at the wavelength of 9.28 μm. The energy percentage of the stray light within the diameter of the Airy disk corresponding to the target wavelength is 18.37% for the incident light with 40 nm bandwidth and 0.1 μm discrete wavelength, and from the point of view of filtering effect, the Q-factor of the integrated metalens reaches 663. In contrast, the energy percentage of the stray light within the diameter of the Airy disk corresponding to the same target wavelength is 72.38% for the metalens without the EIT metasurface under the same conditions.

## 2. Structures and Methods

To design a diffraction limited (DL) hyperbolic metalens, the required phase at position r on the aperture must satisfy Equation (1) [20]:(1)$$\varphi =-\frac{2\pi }{\lambda }\left(\sqrt{r^{2}+f^{2}}-f\right)$$
where *λ* is the operation wavelength and *f* is the target focal length. The reason why a metalens is a diffractive optical element is that meta-atoms with the same structural parameters are selected to meet the identical phase requirement at different radial positions on the aperture of the metalens [20]. The product of the designed focal length and the corresponding working wavelength is a fixed value for the diffractive optical elements, and Equation (2) describes the corresponding mathematical relation [21]:(2)$$f_{1}\cdot \lambda_{1}=f_{2}\cdot \lambda_{2}=\ldots =f_{n}\cdot \lambda_{n}$$
where [*f*_1_, *f*_2_…*f*_n_] is the actual focal length range and [*λ_1_*, *λ*_2_…*λ*_n_] is the corresponding working wavelength range. Equation (2) reveals that the metalens will have a severe chromatic aberration with the increase of the aperture size under a fixed f-number, as described in Equation (3):(3)$$\Delta f=-\frac{\Delta \lambda }{\lambda }\cdot F^{\#}\cdot D$$
where ∆*f* is the variation of the actual focal length, ∆*λ* is the variation of the incident wavelength, *λ* is the operation wavelength, *F^#^* is the f-number, and *D* is the aperture size of the metalens.

To describe the chromatic aberration of the metalens more intuitively, a silicon-based metalens, whose diameter and f-number are 1.00 mm and 1, respectively, is designed to operate at the wavelength of 9.30 μm, and the isolated cylindrical waveguide model is chosen as the physical model of the meta-atoms, as shown in Figure 1a [11]. The height and period of these meta-atoms are 6 μm and 4.75 μm, respectively, and the diameter ranges from 1 to 3 μm. We chose the eight meta-atoms to construct the metalens [11], as shown in Figure 1b.

Based on the eight chosen meta-atoms in Figure 1b, firstly, the focusing property of the metalens with 1 mm diameter and 1 mm focal length is simulated through using the finite difference time domain (FDTD) method. Secondly, the Strehl ratio corresponding to the metalens is calculated as a function of the aperture size for different incident wavelengths [2]. The corresponding results are shown in Figure 1c and Figure 1d, respectively, and it can be seen that the essence of the metalens is a diffractive optical element which has severe chromatic aberration. Therefore, the bandwidth of the metalens decreases significantly with the increase of the aperture size under a fixed f-number, and it is necessary to complement a high Q-factor bandpass filter to improve the imaging performance.

A high Q-factor bandpass filter based on the traditional optical films theory usually means a complex multi-layer optical dielectric films system [22], and the difficulty of preparation process increases significantly with the increase of the aperture size of the metalens due to decreasing of the bandwidth. In contrast, the electromagnetically induced transparency (EIT) metasurface possesses an ability to produce high Q-factor resonances without increasing the complexity of the structure [23,24,25,26]. More appealingly, the same material composition and preparation process as the metalens can be selected for the preparation of the EIT metasurface in the long-wavelength infrared, and the imaging and filtering can be further integrated into a metasurface device. As shown in Figure 2, the germanium (Ge) metasurfaces on both sides of the barium fluoride (BaF_2_) substrate play the role of focusing imaging and filtering, respectively [26,27], and they can be fabricated through reactive-ion etching (RIE) based on the same recipe. The reason for choosing Ge as the base material of the meta-atoms is that its high refractive index (refractive index *n* = 3.96 corresponding to *λ* = 9.3 μm) can support high-quality Mie resonances, and the low refractive index (refractive index *n* = 1.41 corresponding to *λ* = 9.3 μm) of the BaF_2_ substrate is also necessary to retain the original Mie modes of the meta-atoms [28]. Both materials exhibit low optical loss near 0 from 8 to 10 μm to ensure the total efficiency of the integrated metalens [19], and the dispersion of the materials is considered in the simulation process through using discrete data.

The EIT metasurface is usually composed of a broadband bright mode resonator and a narrow band dark mode resonator, and an extremely narrow transmission window, which corresponds to a Fano-type interference, is caused when the two resonances are brought in close proximity in both spatial and frequency domains [25]. As shown in Figure 3, the center wavelength and Q-factor of the narrow passband generated by the EIT metasurface, whose geometrical structure is shown in Figure 3a, can be tailored through adjusting the structure parameters slightly rather than increasing the complexity of the structure, and this feature makes it feasible to design a bandpass filter integrated metalens.

## 3. Results and Discussion

To demonstrate the feasibility of a bandpass filter integrated metalens based on electromagnetically induced transparency, an integrated metalens, whose diameter, f-number, and operation wavelength are 0.30 mm, 0.8, and 9.28 μm, respectively, is simulated. The focusing effect of the integrated metalens with or without EIT metasurface are compared, and the advantages of the integrated metalens are revealed.

Considering the actual situation of the adhesive force of Ge films to BaF_2_ substrate, we chose dielectric Huygens meta-atoms, which possess the characteristics of ultra-thin profile and simple geometrical structure [27,28,29,30], to construct the metalens, and the material composition and structural parameters of the Huygens meta-atoms are shown in Figure 4a. Figure 4b,c shows the simulated amplitude and phase map of the Huygens meta-atoms as a function of the diameter *d* and the incident wavelength *λ*, respectively, and the phase and amplitude of the eight chosen meta-atoms are shown in Figure 4d. The required and realized phase at each radial coordinate on the side of the metalens corresponding to the target wavelength are shown in Figure 4e. On the other side of the integrated metalens, the structural parameters and filtering characteristics of the EIT metasurface are the same as those shown in Figure 3c. Considering the limitation of our computing resources, the thickness of the BaF_2_ substrate was set to 100 μm.

As shown in Figure 2, we chose linear polarization along the x-axis as the polarization mode of the incident electromagnetic wave, and the simulation wavelength range is appropriately broadened. The simulation results in Figure 5a,b show that the introduction of EIT metasurface does not affect the good focusing effect achieved by the metalens at the target wavelength, and greatly reduces the influence of stray light caused by non-target wavelength incident light. The simulation results of the distribution of the light intensity at the focal plane of the target wavelength for different incident wavelengths are shown in Figure 5c,d. The energy percentage of the stray light within the diameter of the Airy disk corresponding to the target wavelength is 18.37% for incident light with 40 nm bandwidth and 0.1 μm discrete wavelength. In contrast, the energy percentage of the stray light within the diameter of the Airy disk corresponding to the same target wavelength is 72.38% for the metalens without the EIT metasurface under the same conditions. As shown in Figure 5e, from the point of view of filtering effect, the Q-factor of the integrated metalens reaches 663. In addition, the existence of the EIT metasurface affects the overall transmittance of the integrated metalens, the focusing efficiency of the metalens for the incident power is reduced from 44.08% to 33.08%.

In addition, for certain wavelength bands, the strategy of the integrated metalens proposed in this paper can be extended by the introduction of active materials to achieve near monochromatic focusing imaging at different positions for different wavelengths through changing the tunable parameters to change the refractive index of the active material [31,32,33].

## 4. Conclusions

In summary, through introducing the EIT metasurface into the design of the LWIR metalens, the imaging performance of the metalens can be further improved. A bandpass filter integrated metalens, whose aperture size and f-number are 0.30 mm and 0.8, respectively, is designed to operate at *λ* = 9.28 μm in this paper; the focusing effect near the diffraction limit can be realized at the target wavelength, and the influence of stray light caused by non-target wavelength incident light can be greatly reduced. The design strategy of a bandpass filter integrated metalens based on electromagnetically induced transparency has certain maneuverability and may have a great potential in the field of LWIR compact optical systems.

## Figures and Tables

**Figure 1 nanomaterials-12-02282-f001:**
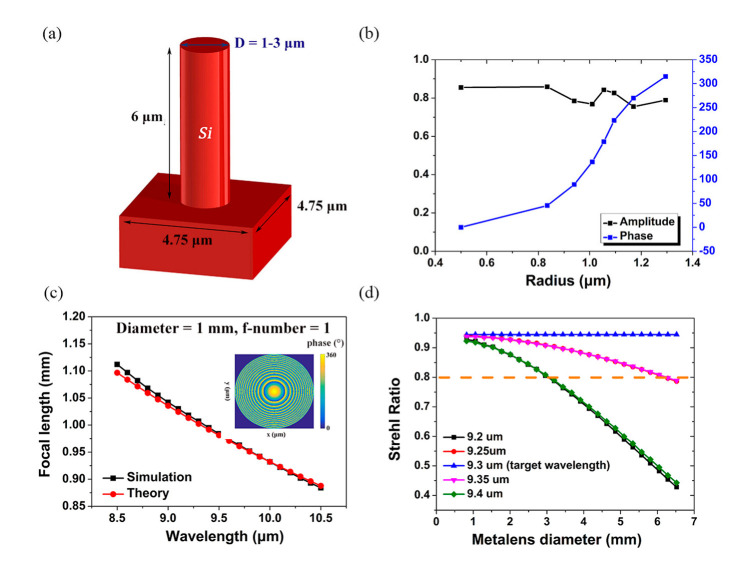
(**a**) The schematic diagram of the isolated cylindrical waveguide model; (**b**) simulated phase and amplitude of the eight chosen meta−atoms at the operation wavelength; (**c**) the variation of actual focal length of the metalens with incident wavelength; (**d**) Strehl ratio of the metalens as a function of the aperture size for different incident wavelengths.

**Figure 2 nanomaterials-12-02282-f002:**
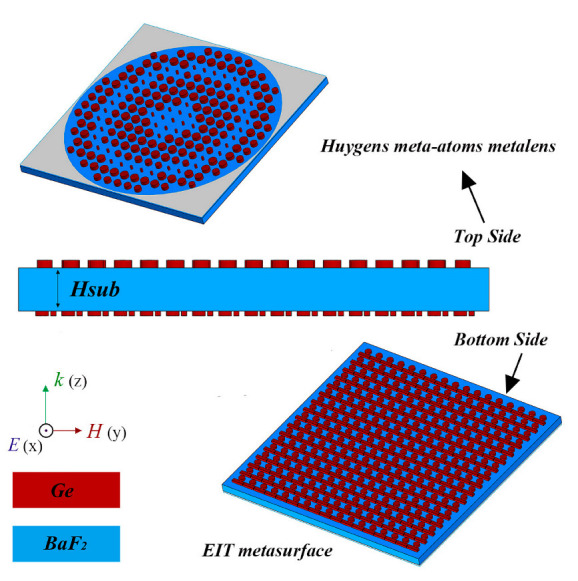
The schematic diagram of a bandpass filter integrated metalens.

**Figure 3 nanomaterials-12-02282-f003:**
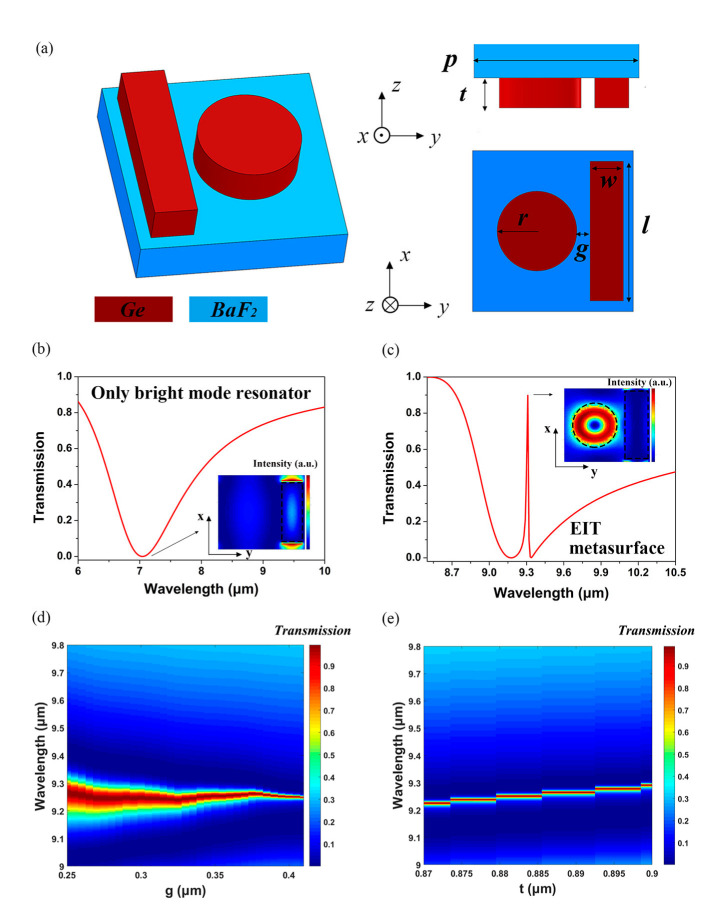
(**a**) The schematic diagram of EIT metasurface; (**b**) the simulated transmittance spectrum of the EIT metasurface without dark mode resonator and the substrate. The inset shows the simulated electric field intensity distribution corresponding to the resonant wavelength. The structure parameters are: *p* = 5.50 μm, *t* = 0.50 μm, *w* = 1.30 μm, *l* = 4.50 μm; (**c**) the simulated transmittance spectrum of the EIT metasurface. The inset shows the simulated electric field intensity distribution corresponding to the transmitted peak. The structure parameters are: *p* = 4.84 μm, *t* = 0.88 μm, *w* = 1.30 μm, *l* = 4.64 μm, *r* = 1.30 μm, and *g* = 0.40 μm; (**d**) the Q-factor of the EIT metasurface as a function of the parameter *g*, and the other structure parameters are the same as those shown in (**c**); (**e**) the transmitted peak of the EIT metasurface as a function of the parameter *t*, and the other structure parameters are the same as those shown in (**c**).

**Figure 4 nanomaterials-12-02282-f004:**
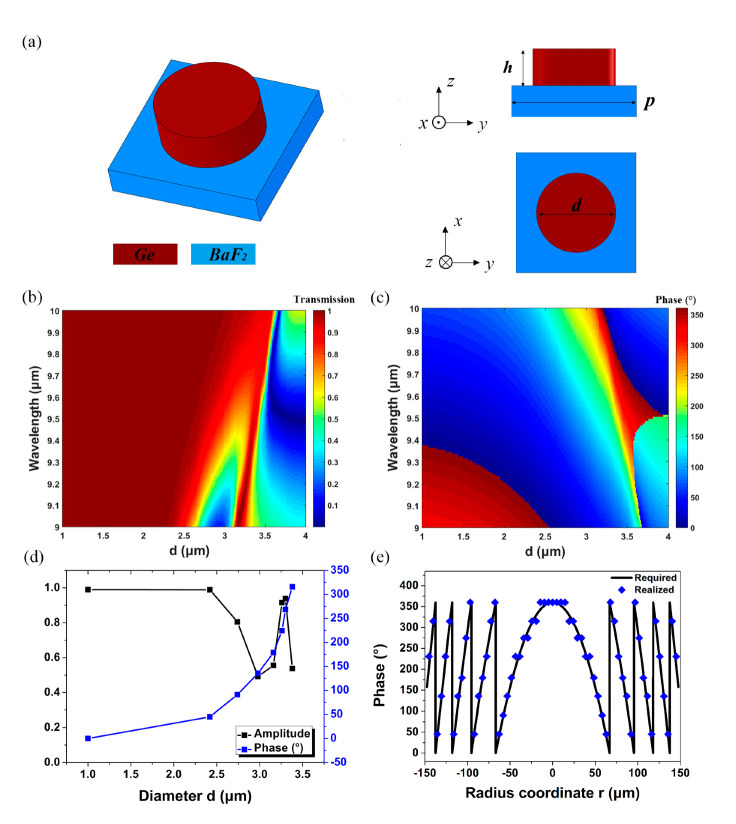
(**a**) The schematic diagram of the dielectric Huygens meta−atoms; (**b**,**c**) the simulated amplitude and phase map of the Huygens meta−atoms as a function of the diameter *d* and the incident wavelength *λ*. The other structure parameters are: *p* = 4.84 μm and *h* = 1.425 μm; (**d**) simulated phase and amplitude of the eight chosen meta−atoms at the operation wavelength. The other structure parameters are: *p* = 4.84 μm and *h* = 1.425 μm; (**e**) required and realized phase at each radial coordinate on the side of the metalens.

**Figure 5 nanomaterials-12-02282-f005:**
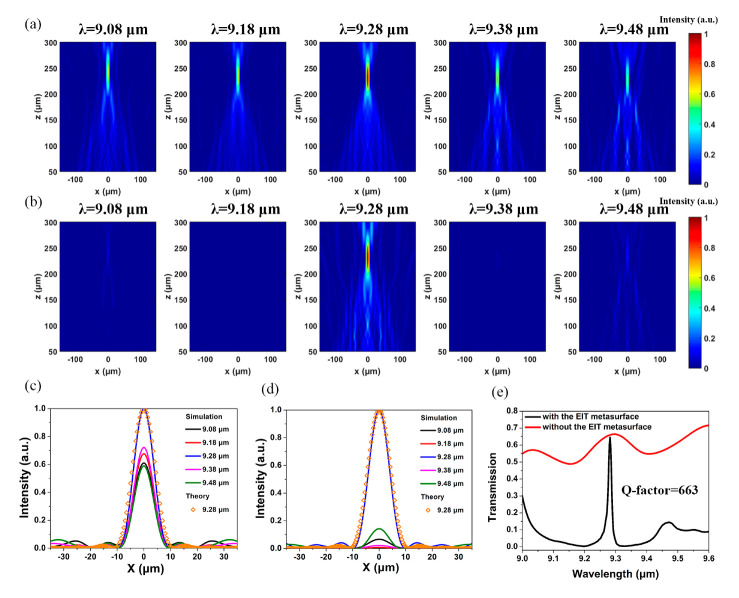
(**a**) The two−dimensional light intensity distribution diagram of the integrated metalens without the EIT metasurface for different wavelengths; (**b**) the two−dimensional light intensity distribution diagrams of the integrated metalens with the EIT metasurface for different wavelengths; (**c**) the one−dimensional light intensity distribution diagram of the integrated metalens without the EIT metasurface for different wavelengths; (**d**) the one−dimensional light intensity distribution diagram of the integrated metalens with the EIT metasurface for different wavelengths; (**e**) the simulated transmittance spectrum of the integrated metalens with or without the EIT metasurface, and the Q−factor corresponding to the integrated metalens with the EIT metasurface reaches 663.

## Data Availability

The data that support the plots within this paper and other findings of this study are available from the corresponding authors on reasonable request.

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
