# Peer review of "Bandpass Filter Integrated Metalens Based on Electromagnetically Induced Transparency"

_nanomaterials, 2022, doi:10.3390/nano12132282_

Round 1
Reviewer 1 Report
Dear Authors:
In Figure 1b name of axis are not indicated.
In rows 97-99 the description of figure 1 is not clear. it seems that in fig1b are reported Strehl ratio e focusing property. please review that paragraph
Author Response
Response to Reviewer 1 Comments
Point 1:
In Figure 1b name of axis are not indicated.
Response 1: Special thanks for your comment. Figure 1b has been revised and the names of axis have been added in the revised manuscript.
Point 2:
In rows 97-99 the description of figure 1 is not clear. it seems that in fig1b are reported Strehl ratio e focusing property. please review that paragraph.
Response 2: Thanks for the Reviewer’s suggestions. The description of figure 1 in rows 97-99 in the manuscript has been revised as follows:
Based on the eight chosen meta-atoms in Figure 1b, firstly, the focusing property of the metalens with 1 mm diameter and 1 mm focal length is simulated through using the finite difference time domain (FDTD) method. Secondly, the Strehl ratio corresponding to the metalens is calculated as a function of the aperture size for different incident wavelengths [2].

Reviewer 2 Report
The authors designed a bandpass filter integrated metalens based on electromagnetically induced transparency (EIT) for long-wavelength infrared (LWIR) imaging. The mechanism of the metadevice has been described in detail, but a few issues require additional discussion:
1. There is no thorough justification of the selected materials, i.e. Ge and BaF2. Why were such materials chosen for the construction of the meta-device? Authors should provide detailed parameters of these materials, i.e. permittivity and losses. Have the authors considered the dispersion of these materials?
2. There is no description of the horizontal axis in Figure 1b.
3. What are the reasons for the differences between the theoretical results and simulations?
4. The device is static and passive, which significantly limits its practical use. Can the device concept be adapted / extended to achieve tunable parameters? In this context, it is worth mentioning the available tuning methods based on liquid crystals, MoS2, graphene, etc. Please see: Applied Physics Letters 102.10 (2013): 102904; Optics express 20.27 (2012): 28664-28671; Applied Physics Letters 102.24 (2013): 241914.
